# Sample Complexity of Posted Pricing for a Single Item

Billy Jin[*]          Thomas Kesselheim[†]          Will Ma[‡]          Sahil Singla[§]

## Abstract

Selling a single item to $n$ self-interested buyers is a fundamental problem in economics, where the two objectives typically considered are welfare maximization and revenue maximization. Since the optimal mechanisms are often impractical and do not work for sequential buyers, posted pricing mechanisms, where fixed prices are set for the item for different buyers, have emerged as a practical and effective alternative. This paper investigates how many samples are needed from buyers' value distributions to find near-optimal posted prices, considering both independent and correlated buyer distributions, and welfare versus revenue maximization. We obtain matching upper and lower bounds (up to logarithmic factors) on the sample complexity for all these settings.

## 1   Introduction

A fundamental problem in Economics is how to sell a single indivisible item to a set of $n$ self-interested buyers. This problem is challenging because the value associated to the item is private knowledge of each buyer and thus can be strategically misreported. The mechanism designer usually has one of two objectives: (a) *welfare* maximization where we want to maximize the value of the winning buyer, and (b) *revenue* maximization where we want to maximize the price paid by the winning buyer. The welfare maximization problem was resolved by Vickrey in 1961 [Vic61] and the revenue maximization problem was resolved (for independent distributions) by Myerson in 1981 [Mye81], both of whom were awarded the Nobel prize in Economics for their contributions. In either case the optimal mechanism is a truthful auction, in which buyers report their values and have no incentive to misreport.

Although we know optimal mechanisms for selling a single item, these auctions are often difficult/impossible to implement in practice. For instance, the buyer does not know exactly how much they will pay if they win the item and these auctions do not work for online settings where the buyers arrive one-by-one [AM+06, HR09]. Thus, the last two decades has seen a lot of progress on understanding the power of simple but near-optimal mechanisms. In this regard, *posted pricing* mechanisms have been identified to be very successful (see books and surveys [Rou16, Har13, CFH+19, Luc17]). Here, the mechanism designer puts a (carefully chosen) price $\pi_i$ on the item and then the $i$-th buyer takes the item if the item is not already sold and if they value it above $\pi_i$. Posted pricing mechanisms have several advantages: they are truthful since $\pi_i$ does not depend on the $i$-th buyer's value; the buyer knows exactly how much they pay on winning an item; and they work even for online settings so the buyers do not have to be simultaneously present. Additionally, in the setting of Myerson's 1981 paper of independent buyer valuations, they are known to give a 2-approximation to both the optimal welfare (which is usually called Prophet Inequality) and the optimal revenue

---

[*]Cornell University, School of Operations Research and Information Engineering, bzj3@cornell.edu.

[†]University of Bonn, Institute of Computer Science and Lamarr Institute for Machine Learning and Artificial Intelligence, thomas.kesselheim@uni-bonn.de

[‡]Columbia University, Graduate School of Business and Data Science Institute, wm2428@gsb.columbia.edu.

[§]Georgia Tech, School of Computer Science, ssingla@gatech.edu. Supported in part by NSF award CCF-2327010.

38th Conference on Neural Information Processing Systems (NeurIPS 2024).

[KS77, KS78, CHK07, CFPV19], as long as the posted prices are set correctly. This raises the main question of the current paper:

> *How many samples are necessary from the value distributions of the buyers to find near-optimal posted prices for single item? Is there a difference between independent vs. correlated distributions or between welfare vs. revenue maximization?*

Despite being a natural question, the tight sample complexity bounds for single item posted pricing are unknown, both for independent/correlated and welfare/revenue settings.

## 1.1 Model

Before stating our results, we formally describe our model.

**Posted pricing.** There is single copy of an item. Buyers $i = 1, \ldots, n$ arrive in order, each with a private value (willingness to pay) $V_i$ for that item. Price $\pi_i$ is offered to buyer $i$, and the first buyer (if any) for whom $V_i \geq \pi_i$ end up buying the item, "winning" the auction. The objective is to maximize either *welfare*, which is the value of the item to the winner, or *revenue*, which is the price paid by the winner.

**Distributions and policies.** We will assume that the vector of values $\mathbf{V} = (V_1, \ldots, V_n)$ is drawn from an unknown but fixed distribution $\mathbf{D}$ over $[0, 1]^n$. If each $V_i$ is drawn *independently* from some marginal distribution $D_i$, then $\mathbf{D}$ is called a *product distribution*, written as $\mathbf{D} = D_1 \times \cdots \times D_n$. Meanwhile, *posted pricing policies* are defined by a vector of prices $\pi = (\pi_1, \ldots, \pi_n)$ that must be fixed before the first buyer arrives, and possibly restricted to some subclass $\Pi$. The objective of policy $\pi$ under buyer values $\mathbf{V}$ is denoted by $\pi(\mathbf{V})$, which equals $V_{\mathrm{argmin}\{i:\pi_i \leq V_i\}}$ and $\pi_{\mathrm{argmin}\{i:\pi_i \leq V_i\}}$ for the welfare and revenue objectives respectively, understood to be $0$ if the item is not sold. We (abusively) let $\pi(\mathbf{D}) := \mathbb{E}_{\mathbf{V} \sim \mathbf{D}}[\pi(\mathbf{V})]$ denote the expected objective of $\pi$, and let $\Pi(\mathbf{D}) := \sup_{\pi \in \Pi} \pi(\mathbf{D})$ denote the best expected objective of a policy in $\Pi$ knowing $\mathbf{D}$.

Note that if we are given the product distribution on buyer values, the optimal policy (for either objective) can be easily computed using dynamic programming [CFPV19]. Consequently, for product distributions we consider the full policy class $\Pi = [0, 1]^n$, and omit the dependence on $\Pi$.

**Learning problem.** A *learning algorithm* takes as input samples $\mathbf{V}$ that are drawn independently and identically distributed (IID) from the unknown distribution $\mathbf{D}$ and an error parameter $\varepsilon \in (0, 1)$, and seeks to return a policy $\pi \in \Pi$ that satisfies $\pi(\mathbf{D}) \geq \Pi(\mathbf{D}) - \varepsilon$, which is called an $\varepsilon$-*approximation*. Given a failure probability $\delta \in (0, 1)$, the *sample complexity* is the minimum number of samples required for there to exist a learning algorithm that, under any distribution $\mathbf{D}$, returns an $\varepsilon$-additive approximation with probability at least $1 - \delta$, noting that the randomness is over both the samples and any random bits in the algorithm. The sample complexity and learning algorithm will depend on the problem variant, defined by the objective (welfare or revenue), any parameters of the policy class $\Pi$, and whether $\mathbf{D}$ is restricted to be a product distribution or not. The sample complexity will also depend on the parameters $\varepsilon, \delta > 0$.

## 1.2 Our Contributions to Posted Pricing for Product Distributions

The study of sample complexity for posted pricing goes back to at least the seminal work of Kleinberg and Leighton [KL03] who study revenue maximization for selling a single item to a single buyer (welfare maximization is trivial for single buyer since we just allocate the item by setting $0$ price). For the general posted pricing problem with $n$ buyers, the best known sample complexity bounds were due to Guo et al. [GHTZ21], who showed that $\tilde{O}(n/\varepsilon^2)$ samples suffice from each buyer's distribution for both welfare and revenue maximization objectives. Although there is a simple $\Omega(1/\varepsilon^2)$ lower bound for both welfare and revenue maximization settings, prior to our work it was unclear if polynomial dependency on the number of buyers $n$ is necessary.

Our first result for product distribution shows that for welfare maximization using posted pricing (a.k.a. prophet inequality), the sample complexity is independent of the number of buyers $n$.

**Theorem 1** (proved in Subsection 2.1)**.** *For product distributions, the sample complexity of welfare maximization is $O(1/\varepsilon^2 \cdot \log^2(1/\delta))$.*

**Proof sketch.** We start by constructing the product empirical distribution using the samples, where for each $i$ we take the uniform distribution on the $\tilde{O}(1/\varepsilon^2)$ samples and then take the product distribution for different $i$. Our main result is that the optimal policy (dynamic programming solution) on this product empirical distribution is an $\varepsilon$-approximation with high probability. Although one could use standard concentration bounds to bound the gap between the learned and optimal thresholds, such arguments lose a poly$(n)$ factor. This is because the difference between successive thresholds is bounded by 1, so naïvely the total variance of $n$ thresholds could be $\Omega(n)$. Our main idea is to instead study a martingale that adds up the errors made in the dynamic programming solution. This way, too high values for one buyer and too low values for another buyer balance each other. On this martingale, we use Freedman's inequality (which is a martingale variant of Bernstein's inequality), which allows us to bound the total variance in terms of conditional variances. A careful application of another concentration bound allows us to bound these conditional variances in terms of the change in the optimal value, whose sum is always at most 1. This latter concentration bound is what causes the additional factor of $\log(1/\delta)$ in the sample complexity.

Our second result for product distribution shows a separation between welfare and revenue maximization using posted pricing, by proving that the $\tilde{O}(n/\varepsilon^2)$ sample-complexity result of [GHTZ21] for revenue maximization is tight up to log factors.

**Theorem 2** (proved in Subsection 2.2). *For revenue maximization on product distributions, any learning algorithm requires $\Omega(\frac{n}{\varepsilon^2})$ samples to return an $\varepsilon$-additive approximation with probability greater than 6/7.*

**Proof sketch.** We construct for each buyer $i$ two possible value distributions that add the same amount (roughly $\frac{1}{n}$) to the value-to-go of the optimal dynamic program. However, these distributions have different optimal prices, and making a mistake (choosing an incorrect price) in isolation loses roughly $\frac{\varepsilon}{n}$ value. Although these mistakes accumulate in a non-linear fashion, we show that making $M$ mistakes must lose in total $\Omega(\frac{\varepsilon M^2}{n^2})$. Finally, these value distributions have probabilities on the scale of $\frac{1\pm\varepsilon}{n}$ with the same supports (essentially, only $1/n$ of samples provide information), which means that $\Omega(\frac{n}{\varepsilon^2})$ samples are needed to avoid making a constant fraction of mistakes.

### 1.3 Our Contributions to Posted Pricing for Correlated Distributions

Independence among buyer valuations can be a strong modeling assumption for many applications. Although for arbitrary correlated distributions the optimal policy is not learnable, one could hope to learn the best policy in the class of all posted pricing policies. A recent work of Balcan et al. [BDD+21] can be applied to this setting to show that $\tilde{O}(n/\varepsilon^2)$ samples are sufficient to learn $\varepsilon$-optimal posted pricing. As discussed in Theorem 2, this linear in $n$ dependency is necessary for revenue maximization, even for product distributions. Our first observation is that for correlated distributions, we need to lose this factor even for welfare maximization.

**Theorem 3** (corollary of Theorem 5). *For welfare maximization with correlated buyers, any learning algorithm requires $\Omega(\frac{n}{\varepsilon^2})$ samples to return an $\varepsilon$-additive approximation with constant probability.*

Given this lower bound, a natural next question is to consider the subclass of posted pricing policies where the algorithm is only allowed to change its threshold at a small number of given locations. Can we now remove the linear in $n$ dependency from sample complexity? The motivation to study this class comes from posted pricing applications where it is not possible for the algorithm designer to update the prices at each time step, e.g., due to business constraints.

Formally, for $\mathcal{S} \subseteq \{2, \ldots, n\}$, we say that posted pricing policy $\pi$ *respects change-points* $\mathcal{S}$ if $\pi_i$ can differ from $\pi_{i-1}$ only when $i \in \mathcal{S}$. We let $\Pi_{\mathcal{S}}$ denote the class of all policies that respect change-points $\mathcal{S}$, noting that policies in $\Pi(\emptyset)$ post a static price for all buyers, and $\Pi(\{2, \ldots, n\}) = [0, 1]^n$. Our following result shows that one can obtain sample complexity that is independent of $n$, depending only on the size of $\mathcal{S}$.

**Theorem 4** (proved in Subsection 3.1). *For correlated distributions, the sample complexity of welfare or revenue maximization is $O\big(\frac{(1+|\mathcal{S}|)\log(1+|\mathcal{S}|)+\log(1/\delta)}{\varepsilon^2}\big)$ when the policy is restricted to $\Pi_{\mathcal{S}}$, for any $\mathcal{S} \subseteq \{2, \ldots, n\}$.*

**Proof sketch.** By existing results in learning theory, it suffices to bound the pseudo-dimension of $\Pi_{\mathcal{S}}$. This will boil down to understanding the structure of "good sets", which are sets of the form $\{\pi \in \Pi_{\mathcal{S}} : \pi(\mathbf{v}) \geq z\}$, for some input $\mathbf{v}$ and some target $z$. We will show that for a natural parameterization of $\Pi_{\mathcal{S}}$, any good set can be expressed as the union and intersection of $O(|\mathcal{S}| + 1)$ halfspaces, which implies the pseudo-dimension of $\Pi_{\mathcal{S}}$ is $O((|\mathcal{S}| + 1) \log(|\mathcal{S}| + 1))$ by a result of [BDD+21]. This bound on the pseudo-dimension translates to the above sample complexity bound.

The learning algorithm which achieves the sample complexity bound in Theorem 4 is simply sample average approximation (SAA), which returns the policy in $\Pi_{\mathcal{S}}$ with the highest objective value averaged over the samples. SAA can be computed in time $O(Tn(Tn)^{1+|\mathcal{S}|})$. This is because there are $1 + |\mathcal{S}|$ prices to decide, each of which can take $Tn$ possible values (one for each realized value in the $T$ samples of length $n$), and evaluating each combination of prices over each of the $T$ samples takes runtime linear in $n$. We leave as an open question whether there is a more efficient algorithm.

Finally, we complement our sample complexity upper bounds for correlated distributions by giving matching lower bounds (up to polylogs).

**Theorem 5** (proved in Appendix A.5). *For welfare or revenue maximization on correlated distributions, a learning algorithm requires $\Omega(\frac{1+|\mathcal{S}|}{\varepsilon^2})$ samples to return an $\varepsilon$-additive approximation with constant probability, when restricted to the policy class $\Pi_{\mathcal{S}}$ for any $\mathcal{S} \subseteq \{2, \ldots, n\}$.*

**Proof sketch.** In the construction for Theorem 5, on each trajectory exactly one of the $1 + |\mathcal{S}|$ decision points (randomly selected) will be relevant, essentially diluting the samples by a factor of $1 + |\mathcal{S}|$ and leading to a lower bound of $\Omega(\frac{1+|\mathcal{S}|}{\varepsilon^2})$.

## 1.4 Further Related Work

In 2007, Hajiaghayi et al. [HKS07] discovered connections between auction design and posted pricing via prophet inequalities. Since then, there is a long line of work on understanding the power of posted pricing for selling multiple items to combinatorial buyers [CMS10, FGL15, DKL20, AKS21, CC23]. For single item revenue maximization with known distributions, Correa et al. [CFPV19] showed the equivalence of welfare and revenue maximization objectives for single item posted pricing.

For background on sample complexity, we suggest Wainwright's excellent textbook [Wai19]. Sample complexity of auction design has been greatly studied; e.g., see [KL03, CR14, MR16]. We refer the readers to Guo et al. [GHZ19], who recently resolved single item revenue maximization in the offline setting, for an overview of the literature. In [GHTZ21], Guo et al. generalized their techniques for revenue maximization over product distributions to all "strongly monotone" problems, which includes posted pricing for welfare and revenue maximization.

Recently, there is also a lot interest in learning auctions in limited feedback models like bandit and pricing queries [GKSW24, SW24, LSTW23]. We should note that sample complexity of optimal stopping (equivalent to our welfare maximization problem) has been previously studied in [GM22], who analyze linear stopping rules in a contextual setting. Our application of techniques from [BDD+21] to online algorithms over correlated sequences is also similar in spirit to some results from [XMX24], who study a different application of inventory optimization.

Another related but tangential line of work focuses on prophet inequalities with samples [AKW14, RWW, CDFS22, CDF+22, DKL+24]. The key distinction in these works is that their benchmark is the expected hindsight optimum, rather than the optimal online policy. Notably, any online algorithm incurs at least a factor of 2 loss compared to the hindsight optimum, even in the case of single-item prophet inequalities. As a result, this line of research aims to achieve $O(1)$-approximation guarantees, rather than the sublinear regret guarantees pursued in the current paper. Furthermore, their techniques differ significantly, as they often assume unbounded distributions. Finally, there is also work that explores the (non-)robustness of algorithms for the prophet inequality problem to inaccuracies in the distributions [DK19] and to dependencies in distributions [ISW20, LPS24].

## 2 Product Distributions

In this section we first prove our improved upper bound on the sample complexity of welfare for product distributions, and then prove a new lower bounds on the sample complexity of revenue for product distributions.

### 2.1 Positive Result for Welfare: Proof of Theorem 1

The reward of the optimal policy is given by the following backward induction: $r_{n+1}^* = 0$ and $r_i^* = \mathbb{E}\left[\max\{r_{i+1}^*, V_i\}\right] = \mathbb{E}\left[(V_i - r_{i+1}^*)^+\right] + r_{i+1}^*$. It sets $\pi_i = r_{i+1}^*$.

When we do not know the distributions but only have $T$ samples from each of them, we can consider the optimal policy on the product empirical distribution, which corresponds to replacing the expectations in the above definitions by the empirical average. That is, we will analyze the policy that sets $\pi_i = \hat{r}_{i+1}$, where $\hat{r}_i$ is defined recursively by $\hat{r}_{n+1} = 0$, $\hat{r}_i = \frac{1}{T}\sum_{t=1}^{T}(V_i^{(t)} - \hat{r}_{i+1})^+ + \hat{r}_{i+1}$.

Let $r_i$ be the expected reward of this policy when starting with the $i$-th arrival. In order to prove the theorem, it is sufficient to show that with probability at least $1 - \delta$, we have $r_1 \geq r_1^* - \epsilon$ if $T \geq (5\ln(2e/\delta)/\epsilon)^2$ for any choices of $\epsilon, \delta \in (0, 1)$.

Let us define $\eta_i = \hat{r}_i - \mathbb{E}\left[\max\{\hat{r}_{i+1}, V_i\}\right]$ or equivalently as

$$\eta_i = \tfrac{1}{T}\sum_{t=1}^{T}(V_i^{(t)} - \hat{r}_{i+1})^+ - \mathbb{E}\left[(V_i - \hat{r}_{i+1})^+\right] \ .$$

That is, $\eta_i$ denotes the error introduced by using the empirical distribution for buyer $i$ instead of the actual one. Note that $\eta_i$ can both be positive and negative.

The two key steps of our proof are as follows. We first show that

$$r_1 \geq r_1^* - 2\max_{j\geq 1}\left|\sum_{i=1}^{j}\eta_i\right| \ . \tag{1}$$

This inequality holds point-wise, that is for any samples drawn. Then, we show that for $T \geq (5\ln(2e/\delta)/\epsilon)^2$ samples, we have $\max_{j\geq 1}\left|\sum_{i=1}^{j}\eta_i\right| \leq \epsilon$ with probability at least $1 - \delta$.

In order to show (1), we first lower bound $r_1$ in terms of $\hat{r}_1$.

**Lemma 1.** $r_1 \geq \hat{r}_1 - \max_{j\in\{0,\dots,n\}}\sum_{i=1}^{j}\eta_i$.

*Proof.* Let $j \in \{0, 1, \dots, n\}$ be the smallest index for which $r_{j+1} \geq \hat{r}_{j+1}$, which exists because $0 = r_{n+1} \geq \hat{r}_{n+1} = 0$. Note that we have $r_i < \hat{r}_i$ for all $1 \leq i \leq j$. We rewrite $r_1$ as

$$r_1 = \mathbf{Pr}\left[V_1 < \hat{r}_2\right]r_2 + \mathbf{Pr}\left[V_1 \geq \hat{r}_2\right]\mathbb{E}\left[V_1 \mid V_1 \geq \hat{r}_2\right] = r_2 + \mathbb{E}\left[(V_1 - r_2)\mathbb{1}_{V_1\geq\hat{r}_2}\right].$$

Repeating this argument,

$$r_1 = \sum_{i=1}^{j-1}\mathbb{E}[(V_i - r_{i+1})\mathbb{1}_{V_i\geq\hat{r}_{i+1}}] + \mathbb{E}[V_j\mathbb{1}_{V_j\geq\hat{r}_{j+1}} + r_{j+1}\mathbb{1}_{V_j<\hat{r}_{j+1}}].$$

Inductively, we can also establish that

$$\hat{r}_1 = \sum_{i=1}^{j-1}\tfrac{1}{T}\sum_t(V_i^{(t)} - \hat{r}_{i+1})^+ + \tfrac{1}{T}\sum_t\max\{V_j^{(t)}, \hat{r}_{j+1}\}.$$

Combining these two equalities,

$$\hat{r}_1 - r_1 = \sum_{i=1}^{j-1}\left(\tfrac{1}{T}\sum_{t=1}^{T}(V_i^{(t)} - \hat{r}_{i+1})^+ - \mathbb{E}[(V_i - \underbrace{r_{i+1}}_{<\hat{r}_{i+1}})\mathbb{1}_{V_i\geq\hat{r}_{i+1}}]\right)$$

$$+ \tfrac{1}{T}\sum_{t=1}^{T}\max\{V_j^{(t)}, \hat{r}_{j+1}\} - \mathbb{E}[V_j\mathbb{1}_{V_j\geq\hat{r}_{j+1}} + \underbrace{r_{j+1}}_{\geq\hat{r}_{j+1}}\mathbb{1}_{V_j<\hat{r}_{j+1}}]$$

$$\leq \sum_{i=1}^{j-1}\left(\tfrac{1}{T}\sum_{t=1}^{T}(V_i^{(t)} - \hat{r}_{i+1})^+ - \mathbb{E}[(V_i - \hat{r}_{i+1})^+]\right) + \tfrac{1}{T}\sum_{t=1}^{T}\max\{V_j^{(t)}, \hat{r}_{j+1}\} - \mathbb{E}[\max\{V_j, \hat{r}_{j+1}\}]$$

$$= \sum_{i=1}^{j}\eta_i.$$

This implies $\hat{r}_1 - r_1 \leq \max_{j\in\{0,\dots,n\}}\sum_{i=1}^{j}\eta_i$, completing the proof. $\square$

A similar proof allows us to prove the following lemma.

**Lemma 2.** $\hat{r}_1 \geq r_1^* - \max_{j \in \{0,\ldots,n\}}(-\sum_{i=1}^j \eta_i)$.

*Proof.* Let $j \in \{0, 1, \ldots, n\}$ be the smallest index for which $\hat{r}_{j+1} \geq r_{j+1}^*$, which exists because $0 = \hat{r}_{n+1} \geq r_{n+1}^* = 0$. Note that we have $\hat{r}_i < r_i^*$ for all $1 \leq i \leq j$, allowing us to derive

$$r_1^* - \hat{r}_1 = \sum_{i=1}^{j-1} \left( \mathbb{E}[(V_i - \underbrace{r_{i+1}^*}_{>\hat{r}_{i+1}})^+] - \frac{1}{T}\sum_{t=1}^T (V_i^{(t)} - \hat{r}_{i+1})^+ \right)$$

$$+ \mathbb{E}[\max\{V_j, \underbrace{r_{j+1}^*}_{\leq \hat{r}_{j+1}}\}] - \frac{1}{T}\sum_{t=1}^T \max\{V_j^{(t)}, \hat{r}_{j+1}\}$$

$$\leq \sum_{i=1}^{j-1} \left( \mathbb{E}[(V_i - \hat{r}_{i+1})^+] - \frac{1}{T}\sum_{t=1}^T (V_i^{(t)} - \hat{r}_{i+1})^+ \right)$$

$$+ \mathbb{E}[\max\{V_j, \hat{r}_{j+1}\}] - \frac{1}{T}\sum_{t=1}^T \max\{V_j^{(t)}, \hat{r}_{j+1}\}$$

$$= -\sum_{i=1}^j \eta_i.$$

This implies $r_1^* - \hat{r}_1 \leq \max_{j \in \{0,\ldots,n\}}(-\sum_{i=1}^j \eta_i)$, completing the proof. $\square$

The last two lemmas imply (1). Thus, we need to bound $\max_j |\sum_{i=1}^j \eta_i|$ to complete the proof.

**Lemma 3.** *For every $\epsilon, \delta > 0$, with probability at least $1 - \delta$, we have $\max_j |\sum_{i=1}^j \eta_i| \leq \epsilon$ if $T \geq (5\ln(2e/\delta)/\epsilon)^2$.*

*Proof.* Observe that

$$\left| \sum_{i=1}^j \eta_i \right| = \left| \sum_{i=1}^n \eta_i - \sum_{i=j+1}^n \eta_i \right| \leq \left| \sum_{i=1}^n \eta_i \right| + \left| \sum_{i=j+1}^n \eta_i \right| \leq 2\max_\tau \left| \sum_{i=\tau}^n \eta_i \right|,$$

so it suffices to show that $\max_\tau |\sum_{i=\tau}^n \eta_i| \geq \epsilon/2$ with probability at most $\delta$. Reversing the quantity of interest to $\max_\tau |\sum_{i=\tau}^n \eta_i|$ allows us to define a martingale, and use Freedman's inequality, which is a martingale version of Bernstein's inequality.

**Lemma 4** (Freedman, Theorem 1.6 in [Fre75]). *Consider a real-valued sequence $\{X_t\}_{t\geq 0}$ such that $X_0 = 0$ and $\mathbb{E}[X_{t+1} \mid X_t, X_{t-1}, \ldots, X_0] = 0$ for all $t$. Assume that the sequence is uniformly bounded, i.e., $|X_t| \leq M$ almost surely for all $t$. Now define the predictable quadratic variation process of the martingale to be $W_t = \sum_{j=0}^t \mathbb{E}[X_j^2 \mid X_{j-1}, \ldots, X_0]$ for all $t \geq 1$. Then for all $\ell \geq 0$ and $\sigma^2 \geq 0$, and any stopping time $\tau$, we have*

$$\mathbf{Pr}\left[ \left| \sum_{j=0}^\tau X_j \right| \geq \ell \text{ and } W_\tau \leq \sigma^2 \right] \leq 2\exp\left( -\frac{\ell^2/2}{\sigma^2 + M\ell/3} \right).$$

A corollary of Freedman's inequality is that $\mathbf{Pr}\left[ \left| \sum_{j=0}^\tau X_j \right| \geq \ell \right] \leq 2\exp\left( -\frac{\ell^2/2}{\sigma^2 + M\ell/3} \right) + \mathbf{Pr}\left[ W_\tau \leq \sigma^2 \right]$. In order to use Freedman's inequality, we consider $nT$ random variables $X_1, \ldots, X_{nT}$, where $X_{iT+t} = \frac{1}{T}\left( (V_{n-i}^{(t)} - \hat{r}_{n-i-1})^+ - \mathbb{E}\left[ (V_{n-i}^{(t)} - \hat{r}_{n-i-1})^+ \right] \right)$ for $i \in \{0, \ldots, n-1\}$ and $t \in \{1, \ldots, T\}$. By this definition,

$$\eta_i = \sum_{t=1}^T X_{(n-i)T+t}$$

because $V_{n-i}^{(t)}$ and $V_{n-i}$ are identically distributed and independent of $\hat{r}_{n-i-1}$. Moreover, for all $j$,

$$\mathbb{E}[X_j \mid X_1, \ldots, X_{j-1}] = 0 \text{ and } |X_j| \leq \frac{1}{T}.$$

Let $\mathcal{E}$ be the event that $\sum_{j=1}^{nT} \mathbb{E}\left[X_j^2 \mid X_1, \ldots, X_{j-1}\right] > \sigma^2 := \frac{e}{e+1} \frac{\ln(2e/\delta)}{T}$. By Freedman's inequality, using $T \geq (5\ln(2e/\delta)/\epsilon)^2$, we have

$$\mathbf{Pr}\left[\max_\tau \left|\sum_{i=\tau}^n \eta_i\right| \geq \tfrac{\epsilon}{2}\right] \leq \mathbf{Pr}\left[\max_\tau \left|\sum_{j=1}^\tau X_j\right| \geq \tfrac{\epsilon}{2}\right]$$

$$\leq 2\exp\left(-\frac{\epsilon^2/8}{\sigma^2 + \frac{\epsilon}{6T}}\right) + \mathbf{Pr}\left[\mathcal{E}\right]$$

$$\leq 2\exp\left(-\frac{\left(\frac{5\ln(2e/\delta)}{\sqrt{T}}\right)^2/8}{\frac{e}{e+1}\frac{\ln(2e/\delta)}{T} + \frac{5\ln(2e/\delta)}{6T\sqrt{T}}}\right) + \mathbf{Pr}\left[\mathcal{E}\right]$$

$$= 2\exp\left(-\frac{25}{\frac{8e}{e+1} + \frac{40}{6\sqrt{T}}}\ln(\frac{2e}{\delta})\right) + \mathbf{Pr}\left[\mathcal{E}\right]$$

$$\leq 2\exp\left(-(1)\ln(\tfrac{4}{\delta})\right) + \mathbf{Pr}\left[\mathcal{E}\right] = \tfrac{\delta}{2} + \mathbf{Pr}\left[\mathcal{E}\right].$$

It remains to show that $\mathbf{Pr}\left[\mathcal{E}\right] \leq \frac{\delta}{2}$. To this end, we observe that for any $i \in \{0, \ldots, n-1\}$ and $t \in \{1, \ldots, T\}$,

$$\mathbb{E}\left[X_{iT+t}^2 \mid X_1, \ldots, X_{iT+t-1}\right]$$
$$= \tfrac{1}{T^2}\mathbb{E}\left[\left((V_{n-i}^{(t)} - \hat{r}_{n-i-1})^+ - \mathbb{E}\left[(V_{n-i}^{(t)} - \hat{r}_{n-i-1})^+\right]\right)^2 \mid X_1, \ldots, X_{iT+t-1}\right]$$
$$\leq \tfrac{1}{T^2}\mathbb{E}\left[(V_{n-i}^{(t)} - \hat{r}_{n-i-1})^+ \mid X_1, \ldots, X_{iT+t-1}\right]$$
$$= \tfrac{1}{T^3}\sum_{t'=1}^T \mathbb{E}\left[(V_{n-i}^{(t')} - \hat{r}_{n-i-1})^+ \mid X_1, \ldots, X_{iT}\right]$$
$$= \frac{1}{T^2}\mathbb{E}\left[\hat{r}_i - \hat{r}_{i+1} \mid \hat{r}_{i+1}, \ldots, \hat{r}_n\right],$$

where the inequality uses $\mathbb{E}\left[Y\right] \geq \mathbb{E}\left[Y^2\right] \geq \mathbb{E}\left[Y^2\right] - (\mathbb{E}\left[Y\right])^2 = \mathbb{E}\left[(Y - \mathbb{E}\left[Y\right])^2\right]$ for any random variable $Y$ with $0 \leq Y \leq 1$ almost surely. In total, we have

$$\sum_{j=1}^{nT}\mathbb{E}\left[X_j^2 \mid X_1, \ldots, X_{j-1}\right] \leq \tfrac{1}{T}\sum_{i=1}^n \mathbb{E}\left[\hat{r}_i - \hat{r}_{i+1} \mid \hat{r}_{i+1}, \ldots, \hat{r}_n\right] .$$

As $\sum_{i=1}^n (\hat{r}_i - \hat{r}_{i+1}) = \hat{r}_1 \leq 1$ almost surely, we have by Lemma 5 (see below) that

$$\sum_{i=1}^n \mathbb{E}\left[\hat{r}_i - \hat{r}_{i+1} \mid \hat{r}_{i+1}, \ldots, \hat{r}_n\right] \geq \frac{e}{e-1}\ln\left(\frac{2e}{\delta}\right)$$

with probability at most $\frac{\delta}{2}$. Therefore, we have

$$\mathbf{Pr}\left[\mathcal{E}\right] \leq \mathbf{Pr}\left[\sum_{i=1}^n \mathbb{E}\left[\hat{r}_i - \hat{r}_{i+1} \mid \hat{r}_{i+1}, \ldots, \hat{r}_n\right] \geq \frac{e}{e-1}\ln\left(\frac{2e}{\delta}\right)\right] \leq \frac{\delta}{2} . \qquad \square$$

**Lemma 5.** *Let $Y_1, Y_2, \ldots, Y_n$ be a sequence of (not necessarily independent) random variables in $[0, 1]$ such that $\sum_{i=1}^n Y_i \leq 1$ almost surely. Then, for any $\delta > 0$, with probability at most $\delta$, we have*

$$\sum_{i=1}^n \mathbb{E}\left[Y_i \mid Y_1, \ldots, Y_{i-1}\right] \geq \frac{e}{e-1}\ln\left(\frac{e}{\delta}\right) .$$

We defer the proof of Lemma 5 to the appendix.

## 2.2 Negative Result for Revenue: Proof of Theorem 2

Each buyer $i = 1, \ldots, n$ has a marginal value distribution that could be $D_i^{\mathrm{H}}$ ("High") or $D_i^{\mathrm{L}}$ ("Low"):

$$D_i^{\mathrm{H}} = \begin{cases} \frac{1}{2} + \frac{n-i}{4n} & \text{w.p. } \frac{1}{2n} \\ \frac{1}{4} + \frac{n-i}{4n} & \text{w.p. } \frac{1}{2n} - 16\frac{\varepsilon}{n} \\ 0 & \text{w.p. } 1 - \frac{1}{n} + 16\frac{\varepsilon}{n} \end{cases} \qquad D_i^{\mathrm{L}} = \begin{cases} \frac{1}{2} + \frac{n-i}{4n} & \text{w.p. } \frac{1}{2n} - 8\frac{\varepsilon}{n} \\ \frac{1}{4} + \frac{n-i}{4n} & \text{w.p. } \frac{1}{2n} + 8\frac{\varepsilon}{n} \\ 0 & \text{w.p. } 1 - \frac{1}{n} \end{cases}$$

which are valid distributions as long as $\varepsilon \leq 1/32$ and $n \geq 2$. All $2^n$ configurations of whether each buyer has the High or Low distribution are possible.

**Optimal policy**  Fix any configuration of whether each buyer has the High or Low distribution. Let $r_i^*$ denote the expected revenue to be earned under the optimal dynamic program, if buyer $i$ is about to arrive and the item is not yet sold. We show inductively that $r_i^* = \frac{n+1-i}{4n}$. By definition $r_{n+1}^* = 0$, establishing $r_i^* = \frac{n+1-i}{4n}$ for $i = n+1$. Now consider $i = n, \ldots, 1$, and assume $r_{i+1}^* = \frac{n-i}{4n}$. Note that it is better to offer one of the prices $\frac{1}{2} + \frac{n-i}{4n}$ or $\frac{1}{4} + \frac{n-i}{4n}$ than to reject the buyer by offering a price of 1, because both of these prices are greater than $r_i^*$ (by the induction hypothesis). Thus, if buyer $i$ has distribution $D_i^{\mathrm{H}}$, then

$$r_i^* = \max\left\{ (\tfrac{1}{2} + \tfrac{n-i}{4n})\tfrac{1}{2n} + r_{i+1}^*(1 - \tfrac{1}{2n}), (\tfrac{1}{4} + \tfrac{n-i}{4n})(\tfrac{1}{n} - 16\tfrac{\varepsilon}{n}) + r_{i+1}^*(1 - \tfrac{1}{n} + 16\tfrac{\varepsilon}{n}) \right\}$$
$$= \max\left\{ \tfrac{1}{2} \cdot \tfrac{1}{2n}, \tfrac{1}{4}(\tfrac{1}{n} - 16\tfrac{\varepsilon}{n}) \right\} + \tfrac{n-i}{4n} \quad = \quad \tfrac{n+1-i}{4n}.$$

Similarly, if buyer $i$ instead has $D_i^{\mathrm{L}}$, then

$$r_i^* = \max\left\{ \tfrac{1}{2}(\tfrac{1}{2n} - 8\tfrac{\varepsilon}{n}), \tfrac{1}{4} \cdot \tfrac{1}{n} \right\} + \tfrac{n-i}{4n} = \tfrac{n+1-i}{4n}.$$

In either case, we have $r_i^* = \frac{n+1-i}{4n}$, completing the induction. Note that $r_1^* = 1/4$.

**Bounding a policy's objective by its number of mistakes**  Now, consider an arbitrary policy $\pi = (\pi_1, \ldots, \pi_n)$ decided by the learning algorithm. Let $r_i$ denote its expected revenue earned if buyer $i$ is about to arrive and the item is not yet sold (under the true configuration of buyer distributions). We can without loss assume $\pi$ to lie in $\{\frac{1}{2} + \frac{n-i}{4n}, \frac{1}{4} + \frac{n-i}{4n}\}^n$, because both prices are higher than $r_{i+1}$, the expected revenue from rejecting (both prices are in fact higher than $r_{i+1}^*$, an upper bound on $r_{i+1}$). We say that the policy *makes a mistake* for buyer $i$ if either $\pi_i = \frac{1}{4} + \frac{n-i}{4n}$ when $i$ has the High distribution, or $\pi_i = \frac{1}{2} + \frac{n-i}{4n}$ when $i$ has the Low distribution. If the policy makes a mistake for $i$, then we have

$$r_i - r_{i+1} \leq \max\left\{ (\tfrac{1}{4} + \tfrac{n-i}{4n} - r_{i+1})(\tfrac{1}{n} - 16\tfrac{\varepsilon}{n}), (\tfrac{1}{2} + \tfrac{n-i}{4n} - r_{i+1})(\tfrac{1}{2n} - 8\tfrac{\varepsilon}{n}) \right\}$$
$$= \tfrac{1}{4n} - 4\tfrac{\varepsilon}{n} + (\tfrac{n-i}{4n} - r_{i+1})(\tfrac{1}{n} - 16\tfrac{\varepsilon}{n});$$

on the other hand, if the policy does not make a mistake for $i$, then we have

$$r_i - r_{i+1} \leq \max\left\{ (\tfrac{1}{2} + \tfrac{n-i}{4n} - r_{i+1})\tfrac{1}{2n}, (\tfrac{1}{4} + \tfrac{n-i}{4n} - r_{i+1})\tfrac{1}{n} \right\} = \tfrac{1}{4n} + (\tfrac{n-i}{4n} - r_{i+1})\tfrac{1}{n}.$$

Hence, if the policy makes $M$ mistakes for some $M \in \{1, \ldots, n\}$, then

$$r_1 = \sum_{i=1}^{n}(r_i - r_{i+1}) \leq \tfrac{1}{4} - M\tfrac{4\varepsilon}{n} + \sum_{i=1}^{n}(\tfrac{n-i}{4n} - r_{i+1})\tfrac{1}{n}.$$

Now, observe that $\frac{n-i}{4n} - r_{i+1} = r_{i+1}^* - r_{i+1} \leq \frac{4\varepsilon}{n}\min\{n-i, M\}$, because the loss from making a mistake is at most $4\frac{\varepsilon}{n}$, and starting from buyer $i + 1$, the number of mistakes can be at most $n - (i+1) + 1 = n - i$ and also at most $M$. Therefore,

$$r_1 \leq \tfrac{1}{4} - M\tfrac{4\varepsilon}{n} + \left((n-M)\tfrac{4\varepsilon}{n}M + \tfrac{4\varepsilon}{n}(M-1) + \cdots + \tfrac{4\varepsilon}{n}\right)\tfrac{1}{n}$$
$$= \tfrac{1}{4} - M\tfrac{4\varepsilon}{n}(1 - \tfrac{n-M}{n} - \tfrac{M-1}{2n}) \ = \ \tfrac{1}{4} - 2\varepsilon\tfrac{M}{n}(\tfrac{M+1}{n}).$$

Recalling that $r_1^* = 1/4$, this shows the additive error is $\Omega(\varepsilon)$ as long as the fraction of mistakes $M/n$ is a constant.

**Computing the Hellinger distance**  We first analyze the Hellinger distance $H(D_i^{\mathrm{H}}, D_i^{\mathrm{L}})$ between (a single observation of) $D_i^{\mathrm{H}}$ vs. $D_i^{\mathrm{L}}$, which we note does not depend on the buyer $i$. The squared Hellinger distance can be bounded using $1 - H^2(D_i^{\mathrm{H}}, D_i^{\mathrm{L}})$

$$= \sqrt{\tfrac{1}{2n}(\tfrac{1}{2n} - 8\tfrac{\varepsilon}{n})} + \sqrt{(\tfrac{1}{2n} + 8\tfrac{\varepsilon}{n} - 24\tfrac{\varepsilon}{n})(\tfrac{1}{2n} + 8\tfrac{\varepsilon}{n})} + \sqrt{(1 - \tfrac{1}{n} + 16\tfrac{\varepsilon}{n})(1 - \tfrac{1}{n})}$$
$$\geq \left(\tfrac{1}{2n} - \tfrac{4\varepsilon}{n} - \tfrac{(8\frac{\varepsilon}{n})^2}{\frac{1}{2n}}\right) + \left(\tfrac{1}{2n} - \tfrac{4\varepsilon}{n} - \tfrac{(24\frac{\varepsilon}{n})^2}{\frac{1}{2n} + \frac{8\varepsilon}{n}}\right) + \left(1 - \tfrac{1}{n} + \tfrac{8\varepsilon}{n} - \tfrac{(16\frac{\varepsilon}{n})^2}{1 - \frac{1}{n}}\right) = 1 - O(\tfrac{\varepsilon^2}{n}),$$

where the inequality applies Lemma 6 below to each square root. This shows that $H^2(D_i^{\mathrm{H}}, D_i^{\mathrm{L}}) = O(\frac{\varepsilon^2}{n})$, and the squared Hellinger distance is additive across independent samples. Using the fact that the Total Variation distance is upper-bounded by $\sqrt{2}$ times the Hellinger distance [GS02], we see that the Total Variation distance between $T$ independent samples of $D_i^{\mathrm{H}}$ vs. $T$ independent samples of $D_i^{\mathrm{L}}$ is $O(\sqrt{\frac{T}{n}}\varepsilon)$. The proof of Lemma 6 is deferred because it is elementary.

**Lemma 6.**  *Suppose $C \in (0, 1)$ and $x \in [-\frac{3}{4}C, \frac{3}{4}C]$. Then $\sqrt{C(C+x)} \geq C + \frac{x}{2} - \frac{x^2}{C}$.*

**Completing the proof of Theorem 2** Suppose the distribution of each buyer is equally likely to be High or Low, independently across buyers. Fix any learning algorithm and the constant probability $1/2$. By the computation of Hellinger distance above, if the number of samples $T$ is less than $C\frac{n}{\varepsilon^2}$ for some constant $C$, then for any buyer $i$, the Total Variation distance between the samples observed under $D_i^{\mathrm{H}}$ vs. $D_i^{\mathrm{L}}$ is at most $1/2$. ($C$ depends on the choice of constant $1/2$.) This means that w.p. at least $1 - 1/2$, the price $\pi_i$ decided by the learning algorithm cannot depend on whether buyer $i$ had distribution $D_i^{\mathrm{H}}$ vs. $D_i^{\mathrm{L}}$, which means that there exists an adversarial choice of $D_i^{\mathrm{H}}$ or $D_i^{\mathrm{L}}$ for each buyer $i$ under which the probability of making a mistake for buyer $i$ is at least $(1 - 1/2)/2 = 1/4$.

Let $M$ denote the (random) number of mistakes, under this adversarial configuration of whether each buyer $i$ has distribution $D_i^{\mathrm{H}}$ or $D_i^{\mathrm{L}}$. We have that $\mathbb{E}[\frac{M}{n}] \geq 1/4$. Although whether the algorithm makes a mistake could be arbitrarily correlated across buyers $i$, applying $\frac{M}{n} \leq 1$, we can employ Markov's inequality on the random variable $1 - \frac{M}{n}$ to see that

$$\mathbb{P}\left[(1 - \tfrac{M}{n}) \geq 7/8\right] \leq \tfrac{3/4}{7/8} = \tfrac{6}{7}.$$

The LHS equals $\mathbb{P}[\frac{M}{n} \leq \frac{1}{8}] = 1 - \mathbb{P}[\frac{M}{n} > \frac{1}{8}]$, and hence $\frac{1}{7} \leq \mathbb{P}[\frac{M}{n} > \frac{1}{8}]$. We have shown that unless $T = \Omega(\frac{n}{\varepsilon^2})$, there is probability at least $1/7$ of making a constant fraction of mistakes, in which case we showed above that the additive error would be $\Omega(\varepsilon)$. This completes the proof of Theorem 2.

## 3 Correlated Distributions

In this section we prove our upper bound on the sample complexity of welfare/revenue maximization for correlated buyer distributions. We show nearly matching lower bounds in Appendix A.5.

### 3.1 Positive Result for Welfare and Revenue: Proof of Theorem 4

To bound the sample complexity of posted pricing for correlated distributions, it suffices to bound the pseudo-dimension of the policy class $\Pi_{\mathcal{S}}$. We use the same approach for both the welfare and revenue objectives. By standard learning theory results [BBL03], bounds on the pseudo-dimension translate to bounds on the sample complexity as follows.

**Theorem 6.** *Let* $\mathrm{PDim}(\Pi_{\mathcal{S}})$ *denote the pseudo-dimension of* $\Pi_{\mathcal{S}}$*. For any* $\epsilon > 0$*, any* $\delta \in (0, 1)$ *and any distribution* $\mathbf{D}$ *over* $[0, 1]^n$*,* $T = O(\frac{1}{\epsilon^2}(\mathrm{PDim}(\Pi_{\mathcal{S}}) + \log\frac{1}{\delta}))$ *samples are sufficient to ensure that with probability at least* $1 - \delta$ *over the draw of samples* $(\mathbf{v}_1, \ldots, \mathbf{v}_T) \sim \mathbf{D}^T$*, for all* $\pi \in \Pi_{\mathcal{S}}$*,*

$$\left|\tfrac{1}{T}\sum_{t=1}^{T}\pi(\mathbf{v}_t) - \pi(\mathbf{D})\right| \leq \epsilon.$$

We will show that $\mathrm{PDim}(\Pi_{\mathcal{S}}) = O(k\log k)$, where $k = |\mathcal{S}| + 1$. This together with the above theorem immediately implies that Theorem 4 holds for the sample average approximation algorithm.

Let $\mathcal{S} = \{i_1, i_2, i_3, \ldots, i_{k-1}\}$, where $1 < i_1 < i_2 < \ldots < i_{k-1}$. For each $j = 1, 2, \ldots, k$, let $I_j = \{i_{j-1}, \ldots, i_j - 1\}$, with the convention that $i_0 = 1$ and $i_k = n + 1$. In other words, $I_1, I_2, \ldots, I_k$ are consecutive intervals that partition $[n]$, and $\Pi_{\mathcal{S}}$ is the class of policies that offers every customer in $I_j$ the same price. Note that every policy $\pi \in \Pi_{\mathcal{S}}$ can be parameterized by $k$ prices $\rho = (\rho_1, \rho_2, \ldots, \rho_k)$, where $\rho_j$ is the price offered to customers in $I_j$. In the rest of this proof, we will use $\pi_\rho$ to denote the policy in $\Pi_{\mathcal{S}}$ parameterized by $\rho$.

By the definition of pseudo-dimension,

$$\mathrm{PDim}(\Pi_{\mathcal{S}}) = \mathrm{VCdim}(\widetilde{\Pi}_{\mathcal{S}}), \tag{2}$$

where $\widetilde{\Pi}_{\mathcal{S}} := \{(\mathbf{v}, z) \mapsto \mathbb{1}\{\pi_\rho(\mathbf{v}) \geq z\} : \rho \in \mathbb{R}^k\}$. Let $\widetilde{\Pi}_{\mathcal{S}}^*$ denote the dual class of $\widetilde{\Pi}_{\mathcal{S}}$, so

$$\widetilde{\Pi}_{\mathcal{S}}^* := \{\rho \mapsto \mathbb{1}\{\pi_\rho(\mathbf{v}) \geq z\} : \mathbf{v} \in [0, 1]^n, z \in \mathbb{R}\}.$$

We will use a result from [BDD+21] to bound the VC dimension of $\widetilde{\Pi}_{\mathcal{S}}$. We state this result below in Definition 1 and Theorem 7. This result essentially says that the pseudo-dimension of the primal class is bounded if the dual class is well-structured. Here, "well-structured" essentially means that the domain can be partitioned into pieces defined by a small number of boundary functions, and the function is simple on each piece.

**Definition 1** (Definition 3.2 in [BDD+21]). *A function class $\mathcal{H} \subseteq \mathbb{R}^{\mathcal{Y}}$ that maps a domain $\mathcal{Y}$ to $\mathbb{R}$ is $(\mathcal{F}, \mathcal{G}, l)$-piecewise decomposable for a class $\mathcal{G} \subseteq \{0,1\}^{\mathcal{Y}}$ of boundary functions and a class $\mathcal{F} \subseteq \mathbb{R}^{\mathcal{Y}}$ of piece functions if the following holds: for every $h \in \mathcal{H}$, there are $l$ boundary functions $g^{(1)}, \ldots, g^{(l)} \in \mathcal{G}$ and a piece function $f_b \in \mathcal{F}$ for each bit vector $b \in \{0,1\}^l$ such that for all $y \in \mathcal{Y}, h(y) = f_{b_y}(y)$ where $b_y = (g^{(1)}(y), \ldots, g^{(l)}(y)) \in \{0,1\}^l$.*

The main theorem in [BDD+21] states that if the dual class is $(\mathcal{F}, \mathcal{G}, l)$-piecewise decomposable, then the pseudo-dimension of the primal class is bounded.

**Theorem 7** (Theorem 3.3 in [BDD+21]). *Suppose that the dual function class $\mathcal{U}^*$ is $(\mathcal{F}, \mathcal{G}, l)$-piecewise decomposable with boundary functions $\mathcal{G} \subseteq \{0,1\}^{\mathcal{U}}$ and piece functions $\mathcal{F} \subseteq \mathbb{R}^{\mathcal{U}}$. The pseudo-dimension of $\mathcal{U}$ is bounded as follows:*

$$\mathrm{PDim}(\mathcal{U}) = O((\mathrm{PDim}(\mathcal{F}^*) + \mathrm{VCdim}(\mathcal{G}^*))\ln(\mathrm{PDim}(\mathcal{F}^*) + \mathrm{VCdim}(\mathcal{G}^*)) + \mathrm{VCdim}(\mathcal{G}^*)\ln l).$$

We now apply Theorem 7 to bound the VC dimension of $\widetilde{\Pi}_{\mathcal{S}}$.[5] To do so we must show that the dual class $\widetilde{\Pi}_{\mathcal{S}}^*$ is $(\mathcal{F}, \mathcal{G}, l)$-piecewise decomposable for "nice" classes $\mathcal{F}$ and $\mathcal{G}$. We will show that for both the welfare and revenue objectives, we can take $\mathcal{F}$ to be the family of constant functions and $\mathcal{G}$ to be the family of axis-aligned halfspaces. This will follow from the below lemma, whose proof is deferred to the Appendix for space reasons.

**Lemma 7.** *Let $\mathbf{v} \in [0,1]^n$ and let $z \in \mathbb{R}$. Let $G(\mathbf{v}, z) = \{\rho \in \mathbb{R}^k : \pi_\rho(\mathbf{v}) \geq z\}$. For both the welfare and revenue objectives, there are $l_1, u_1, \cdots, l_k, u_k \in \mathbb{R} \cup \{\pm\infty\}$ such that*

$$G(\mathbf{v}, z) = \bigcup_{j=1}^{k} (u_1, \infty) \times \cdots \times (u_{j-1}, \infty) \times (l_j, u_j] \times \mathbb{R}^{k-j}. \tag{3}$$

**Corollary 1.** *$\widetilde{\Pi}_{\mathcal{S}}^*$ is $(\mathcal{F}, \mathcal{G}, l)$-piecewise decomposable, where $\mathcal{F}$ is the set of constant functions, $\mathcal{G}$ is the set of axis-aligned halfspaces, and $l = 2(|\mathcal{S}| + 1)$.*

*Proof.* Consider a function $h \in \widetilde{\Pi}_{\mathcal{S}}^*$. By definition of $\widetilde{\Pi}_{\mathcal{S}}^*$, there exist $\mathbf{v} \in [0,1]^n$ and $z \in \mathbb{R}$ such that $h(\rho) = \mathbb{1}\{\pi_\rho(\mathbf{v}) \geq z\}$. By Lemma 7, there are $l_1, u_1, \ldots, l_k, u_k$ such that

$$\{\rho \in \mathbb{R}^k : h(\rho) = 1\} = \bigcup_{j=1}^{k} (u_1, \infty) \times \cdots \times (u_{j-1}, \infty) \times (l_j, u_j] \times \mathbb{R}^{k-j}.$$

For each $j \in [k]$, let $L_j = \{\rho \in \mathbb{R}^k : \rho_j > l_j\}$ and $U_j = \{\rho \in \mathbb{R}^k : \rho_j \leq u_j\}$. Note that $L_j$ and $U_j$ are axis-aligned halfspaces, and

$$\{\rho \in \mathbb{R}^k : h(\rho) = 1\} = \bigcup_{j=1}^{k} \bar{U}_1 \cap \bar{U}_2 \cap \cdots \cap \bar{U}_{j-1} \cap L_j \cap U_j.$$

Therefore, in the definition of $(\mathcal{F}, \mathcal{G}, l)$-piecewise decomposable, we may take the boundary functions to be the $2k$ functions corresponding to the halfspaces $L_1, U_1, \ldots, L_k, U_k$. On any given piece defined by these boundary functions, $h$ is a constant function (equal to either 0 or 1). $\square$

For $\mathcal{F}$ the set of constant functions and $\mathcal{G}$ the set of axis-aligned halfspaces in $\mathbb{R}^k$, it is easy to check that $\mathrm{PDim}(\mathcal{F}^*) = 0$ and $\mathrm{VCdim}(\mathcal{G}^*) = k$. Combining Corollary 1 with Theorem 7, we get that

$$\mathrm{VCdim}(\widetilde{\Pi}_{\mathcal{S}}) = O\left(k \ln k + k \ln 2k\right) = O(k \ln k).$$

This completes the proof of Theorem 4.

**Remark.** If we directly apply Theorem 7 from [BDD+21] to bound $\mathrm{PDim}(\Pi_{\mathcal{S}})$, then we would get

- A $O(k \ln k)$ pseudo-dimension bound for the revenue objective;
- A $O(k \ln(kn))$ pseudo-dimension bound for welfare objective.

In particular, the bound for welfare would grow with $n$. This dependence on $n$ is in fact unavoidable if one works with $\Pi_{\mathcal{S}}$ directly, because for instances like $\mathbf{v} = (\frac{1}{n}, \frac{2}{n}, \ldots, 1)$, the dual function corresponding to $\mathbf{v}$ is piecewise constant with $\Theta(n)$ pieces, even if $k = 1$. This is why our Corollary 1 focuses on the indicator functions $\widetilde{\Pi}_{\mathcal{S}}^*$ instead. Regardless, both the bounds for welfare and revenue require our Lemma 7 that analyzes the problem-specific structure of the "good sets".

---

[5]Note Theorem 7 is stated to bound the pseudo-dimension, but the pseudo-dimension coincides with the VC dimension for function classes consisting of $\{0,1\}$-valued functions.

**Acknowledgement**    This work was done in part while the authors were visiting the Simons Institute for the Theory of Computing for the program on Data-Driven Decision Processes. The authors thank Zhuoxin Chen for identifying typos in an early version.

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

# A   Appendix

## A.1   Proof of Lemma 2

*Proof.* Now, consider the smallest $j$ so that $\hat{r}_j \geq r_j^*$. Again, we know such a $j$ exists because $0 = \hat{r}_{n+1} \geq r_{n+1}^* = 0$. We claim that that $\hat{r}_1 \geq r_1^* + \min\{\sum_{i=1}^{j-1} \eta_i, 0\}$.

Observe that there is nothing to be shown if $j = 1$ because then $\hat{r}_1 \geq r_1^*$. So, let's consider the case that $j > 1$. We can now write $\hat{r}_1$ as a telescoping sum

$$\hat{r}_1 = \hat{r}_{j-1} + \sum_{i=1}^{j-2}(\hat{r}_i - \hat{r}_{i+1}) \ . \tag{4}$$

Since $\hat{r}_j \geq r_j^*$, we have

$$\begin{aligned}
\hat{r}_{j-1} &= \hat{r}_j + \tfrac{1}{T}\sum_{t=1}^{T}(V_{j-1}^{(t)} - \hat{r}_j)^+ = \hat{r}_j + \mathbb{E}\left[(V_{j-1} - \hat{r}_j)^+\right] + \eta_{j-1} \\
&= \mathbb{E}\left[\max\{V_{j-1}, \hat{r}_j\}\right] + \eta_{j-1} \geq \mathbb{E}\left[\max\{V_{j-1}, r_j^*\}\right] + \eta_{j-1} = r_{j-1}^* + \eta_{j-1} \ .
\end{aligned}$$

Observe that for $i + 1 < j$, we have $\hat{r}_{i+1} < r_{i+1}^*$. Therefore

$$\hat{r}_i - \hat{r}_{i+1} = \tfrac{1}{T}\sum_{t=1}^{T}(V_i^{(t)} - \hat{r}_{i+1})^+ = \mathbb{E}\left[(V_i - \hat{r}_{i+1})^+\right] + \eta_i \geq \mathbb{E}\left[(V_i - r_{i+1}^*)^+\right] + \eta_i = r_i^* - r_{i+1}^* + \eta_i \ .$$

So, in combination with (4),

$$\hat{r}_1 \geq r_{j-1}^* + \eta_{j-1} + \sum_{i=1}^{j-2}(r_i^* - r_{i+1}^* + \eta_i) = r_1^* + \sum_{i=1}^{j-1} \eta_i \ . \qquad \square$$

## A.2   Proof of Lemma 5

Ignoring constants, a slick proof of this lemma is via another application of Freedman's inequality, where we define a martingale $X_i = Y_i - \mathbb{E}[Y_i \mid Y_1, \ldots, Y_{i-1}]$, and cut this martingale (stop) whenever the sum of conditional variances exceeds $\log(1/\delta)$. This bounds the sum of conditional variances by $\log(1/\delta)$, i.e. $W_\tau \leq \sigma^2 := \log(1/\delta)$ w.p. 1. Since the lemma is upper-bounding the probability of the event that $\sum_i X_i > \log(1/\delta)$, we can show that this probability is $O(\delta)$ by substituting $\ell = \log(1/\delta)$ into Freedman's inequality.

Below we give another, elementary proof of this lemma.

*Proof.* Let $Z = 1$ if $\sum_{i=1}^{n} \mathbb{E}[Y_i \mid Y_1, \ldots, Y_{i-1}] \geq \frac{e}{e-1}\ln\left(\frac{e}{\delta}\right)$. By Markov's inequality, we have

$$\mathbf{Pr}\left[\sum_{i=1}^{n} Y_i \leq 1 \text{ and } Z = 1\right] = \mathbf{Pr}\left[Z e^{-\sum_{i=1}^{n} Y_i} \geq e^{-1}\right] \leq \mathbb{E}\left[Z e^{-\sum_{i=1}^{n} Y_i}\right] \cdot e$$

Now, we have

$$\mathbb{E}\left[Z e^{-\sum_{i=1}^{n} Y_i}\right] = \mathbb{E}\left[Z \prod_{i=1}^{n} e^{-Y_i}\right] = \mathbb{E}\left[\mathbb{E}[Z \mid Y_1, \ldots, Y_n] \prod_{i=1}^{n} \mathbb{E}\left[e^{-Y_i} \mid Y_1, \ldots, Y_{i-1}\right]\right]$$

For every single conditional expectation, we obtain by convexity

$$\begin{aligned}
\mathbb{E}\left[e^{-Y_i} \mid Y_1, \ldots, Y_{i-1}\right] &\leq \mathbb{E}\left[Y_i e^{-1} + (1 - Y_i) \mid Y_1, \ldots, Y_{i-1}\right] \\
&= \mathbb{E}\left[Y_i(e^{-1} - 1) \mid Y_1, \ldots, Y_{i-1}\right] + 1 \\
&\leq \exp\left(\mathbb{E}[Y_i \mid Y_1, \ldots, Y_{i-1}](e^{-1} - 1)\right) \ .
\end{aligned}$$

Now, consider a fixed realization of $Y_1, \ldots, Y_n$. This realization fully determines $Z$. If $Z = 0$, then

$$\mathbb{E}[Z \mid Y_1, \ldots, Y_n] \prod_{i=1}^{n} \mathbb{E}\left[e^{-Y_i} \mid Y_1, \ldots, Y_{i-1}\right] = 0 \ .$$

Otherwise, with $Z = 1$, we have

$$\mathbb{E}[Z \mid Y_1, \ldots, Y_n] \prod_{i=1}^{n} \mathbb{E}\left[e^{-Y_i} \mid Y_1, \ldots, Y_{i-1}\right] \leq \exp\left(\sum_{i=1}^{n} \mathbb{E}[Y_i \mid Y_1, \ldots, Y_{i-1}](e^{-1} - 1)\right) \leq \frac{\delta}{e} \ .$$

So, we have an upper bound of $\frac{\delta}{e}$ regardless of the realization. Taking an expectation, we get

$$\mathbb{E}\left[\mathbb{E}\left[Z \mid Y_1, \ldots, Y_n\right] \prod_{i=1}^{n} \mathbb{E}\left[e^{-Y_i} \mid Y_1, \ldots, Y_{i-1}\right]\right] \leq \frac{\delta}{e},$$

which implies $\mathbf{Pr}\left[\sum_{i=1}^{n} Y_i \leq 1 \text{ and } Z = 1\right] \leq \mathbb{E}\left[Z e^{-\sum_{i=1}^{n} Y_i}\right] \cdot e \leq \frac{\delta}{e} e = \delta$, completing the proof. $\qquad\square$

## A.3 Proof of Lemma 6.

*Proof.* Let $f(x) = \sqrt{C(C + x)} = (C^2 + Cx)^{1/2}$, which is a continuous function over $x \in [-\frac{3}{4}C, \frac{3}{4}C]$. We have $f'(x) = \frac{1}{2}(C^2 + Cx)^{-1/2}C$ and $f''(x) = -\frac{1}{4}(C^2 + Cx)^{-3/2}C^2$, both of which exist over $x \in [-\frac{3}{4}C, \frac{3}{4}C]$. Applying Taylor's theorem around 0, we get

$$
\begin{aligned}
f(x) = f(0) + f'(0)x + \tfrac{f''(y)}{2}x^2 &= C + \frac{1}{2}x - \frac{1}{8}(C^2 + Cy)^{-3/2}C^2 x^2 \\
&\geq C + \tfrac{x}{2} - \tfrac{1}{8}(C^2 + C(-\tfrac{3}{4}C))^{-3/2}C^2 x^2 = C + \tfrac{x}{2} - \tfrac{x^2}{C},
\end{aligned}
$$

where $y$ lies between 0 and $x$ and the inequality holds because $y \geq -\frac{3}{4}C$, completing the proof. $\quad\square$

## A.4 Proof of Lemma 7

If $z \leq 0$ then $G(\mathbf{v}, z) = \mathbb{R}^k$, so we may take any values of $l_1, u_1, \ldots, l_k, u_k$ such that $l_1 = l_2 = \cdots = l_k - \infty$ and $u_k = \infty$.

For the rest of the proof, assume $z > 0$. First, consider the welfare objective. We show the lemma holds for $l_j$ and $u_j$ as follows:

Define $l_j$ to be $-\infty$ if $\mathbf{v}(I_j)_1 \geq z$, and otherwise to be $\max\{\mathbf{v}(I_j)_1, \ldots, \mathbf{v}(I_j)_{m-1}\}$, where $m$ is the smallest index such that $\mathbf{v}(I_j)_m \geq z$. (If $\mathbf{v}(I_j)_m < z$ for all $m$, set $l_j = \max(\mathbf{v}(I_j))$.)

Define $u_j$ to be $\max(\mathbf{v}(I_j))$.

Let $G = \{\rho \in \mathbb{R}^k : \pi_\rho(\mathbf{v}) \geq z\}$ and $G' = \bigcup_{j=1}^{k}(u_1, \infty) \times \cdots \times (u_{j-1}, \infty) \times (l_j, u_j] \times \mathbb{R}^{k-j}$. To show that $G = G'$, we'll show $G \subseteq G'$ and $G' \subseteq G$.

Case 1 ($G \subseteq G'$). If $G = \emptyset$ we are done, so assume otherwise. Let $\rho \in G$. Let $j$ be the interval such that the algorithm (using thresholds $\rho$) accepts a value in $\mathbf{v}(I_j)$. Since a value in interval $j$ was accepted, $\rho_j \leq \max(\mathbf{v}(I_j)) = u_j$. Also, since no value in the previous intervals were accepted, $\rho_i > u_i$ for all $i < j$. Finally, we must have $\rho_j > l_j$, since otherwise the accepted value will be less than $z$. Thus $\rho \in G'$.

Case 2 ($G' \subseteq G$). Let $\rho \in G'$. Then $\rho \in (u_1, \infty) \times \cdots \times (u_{j-1}, \infty) \times (l_j, u_j] \times \mathbb{R}^{k-j}$ for some $j$. Since $\rho_i > u_i$ for all $i < j$, the algorithm using thresholds $\rho$ does not accept any value in the intervals $i < j$. Since $\rho_j \in (l_j, u_j]$, the definitions of $l_j$ and $u_j$ imply that a value in $\mathbf{v}(I_j)$ is accepted, and this value is greater than or equal to $z$. Thus $\rho \in G$.

For revenue, the only difference is that we define $l_j$ to be $\min(z, \max(\mathbf{v}(I_j)))$. ($u_j$ is still defined to be $\max(\mathbf{v}(I_j))$.) With these definitions of $l_j$ and $u_j$, a very similar analysis shows that $G = G'$ in the revenue case as well.

## A.5 Proof of Theorem 5

We let $\mathcal{S}' \subseteq \{1\} \cup \mathcal{S}$ be a subset of decision points such that price $\pi_i$ can be freely chosen for all $i \in \mathcal{S}'$. For welfare maximization we require $i + 1 \in \{1, \ldots, n\} \setminus \mathcal{S}'$ for all $i \in \mathcal{S}'$, which can achieved while ensuring $|\mathcal{S}'| \geq \lfloor\frac{1+|\mathcal{S}|}{2}\rfloor$. Each decision point $i \in \mathcal{S}$ has marginal value distribution(s) that could be "High" or "Low", separately for each decision point (i.e., all $2^{|\mathcal{S}'|}$ combinations are possible). For welfare maximization, they are defined as

- High: $V_i = 1/2$ with probability (w.p.) 1, $V_{i+1} = 1$ w.p. $1/2 + \varepsilon$, $V_{i+1} = 0$ w.p. $1/2 - \varepsilon$;
- Low: $V_i = 1/2$ w.p. 1, $V_{i+1} = 1$ w.p. $1/2 - \varepsilon$, $V_{i+1} = 0$ w.p. $1/2 + \varepsilon$.

For revenue maximization, they are defined as

- High: $V_i = 1$ w.p. $1/2 + \varepsilon$, $V_i = 1/2$ w.p. $1/2 - \varepsilon$;
- Low: $V_i = 1$ w.p. $1/2 - \varepsilon$, $V_i = 1/2$ w.p. $1/2 + \varepsilon$.

The overall distribution over trajectories $\mathbf{V}$ is then correlated as follows. First, a decision point $\tilde{i} \in \mathcal{S}'$ is drawn uniformly at random. Then, $V_{\tilde{i}}$ (as well as $V_{\tilde{i}+1}$ in the case of welfare maximization) is drawn according to the distributions above, depending on whether decision point $\tilde{i}$ is High or Low. All other buyer values are 0 on this trajectory.

Given this, only the prices for decision point $\tilde{i}$ are relevant, which can without loss be restricted to $\{1/2, 1\}$. The policy should optimize $\pi_i$ for each $i \in \mathcal{S}'$ as if $\tilde{i} = i$. For either welfare or revenue maximization, the constructions above can be checked to satisfy the following properties:

1. Setting $\pi_i$ (and $\pi_{i+1}$ in the case of welfare maximization) to 1 is optimal for High and earns objective $1/2 + \varepsilon$ in expectation; setting to $1/2$ is optimal for Low and earns objective $1/2$;

2. Setting the wrong price earns expected objective $1/2$ for High and $1/2 - \varepsilon$ for Low, incurring a loss of $\varepsilon$ compared to optimal in both cases.

3. The High and Low distributions have the same support and probabilities that differ by $2\varepsilon$.

We note that for welfare maximization, it does not matter whether $\pi_{i+1} \in \mathcal{S}$, because the decision is on whether to accept buyer $i$ when $V_i = 1/2$.

Finally, a policy has additive error $\Omega(\varepsilon)$ as long as it has constant probability of setting the wrong price for a decision point. Due to property 3. above, the policy needs $\Omega(\frac{1}{\varepsilon^2})$ relevant observations for a given decision point to avoid setting the wrong price for it with constant probability. However, each sample contains a relevant observation for a given decision point only with probability $1/|\mathcal{S}'|$, so $\Omega(\frac{|\mathcal{S}'|}{\varepsilon^2})$ samples are necessary for the number of relevant observations to be $\Omega(\frac{1}{\varepsilon^2})$ with high probability. Because $|\mathcal{S}'| \geq \lfloor \frac{1+|\mathcal{S}|}{2} \rfloor$, this completes the proof of Theorem 5.

