# OpenReview forum: "Sample Complexity of Posted Pricing for a Single Item"
_NeurIPS.cc/2024/Conference — NeurIPS 2024 spotlight_

### Official Review · Reviewer_L1MB · 2024-07-10

**Soundness:** 4
**Presentation:** 4
**Contribution:** 4
**Rating:** 8
**Confidence:** 4

**Summary:**

This paper studies the sample complexity of posted pricing problems. In such problems, a sequence of $n$ buyers arrive with valuations drawn from fixed distributions, and the goal is to post prices that maximize either the revenue or the welfare. In particular, the item is sold to the first buyer who accepts its proposed price. Notably, in the welfare maximizing case, this is the prophet inequality problem.

Given access to the distributions of the buyers’ valuations, it is possible to find the optimal online strategy via dynamic programming. This paper studies this problem when the access to the distribution is given via samples. The authors provide matching sample complexity bounds for the revenue and welfare maximization tasks, and for both the independent and correlated setting.

**Strengths:**

- Posted pricing is an important problem, vastly studied in both the EC and NeurIPS/ICML communities, with countless applications
- Studying its sample complexity is well-motivated
- the paper significantly improves on the state-of-the-art [Guo et al. COLT’21], removing the dependence on $n$
- The paper is well-written

**Weaknesses:**

The paper is a strong NeurIPS submission

**Questions:**

None

**Limitations:**

No potential negative societal impact of this work

---

> ### Author Rebuttal · Authors · 2024-08-06
>
> Thank you for your thoughtful review!

---

### Official Review · Reviewer_U8bp · 2024-07-15

**Soundness:** 4
**Presentation:** 3
**Contribution:** 2
**Rating:** 6
**Confidence:** 4

**Summary:**

This paper studies the problem of learning approximately revenue (welfare) optimal posted-prices single-item auctions from samples. The authors consider the setting where the valuation distributions of the agents are independent and the setting where the distributions are correlated. For both settings and both objectives they derive (almost) matching lower and upper bounds on the sample complexity of the learning task. All the results restrict the valuation distributions to be bounded in $[0,1]$ and consider additive approximations of the optimal posted-prices auctions.

**Strengths:**

-The problem of learning posted-prices auctions is of interest to the community.

-The results derived are almost tight.

-The technical results require some work (even though, conceptually, the approach relies heavily on prior work).

**Weaknesses:**

-The conceptual ideas of the work feel a bit incremental, given the long line of work in the setting of learning optimal auctions.

-The results are limited to a very simple setting of single item auctions.

-The running time of the algorithm for correlated distributions is exponential.

-The presentation could be improved a bit. For example, I found the use of the word "policy" a bit confusing. Usually, this would imply some dynamic decision process, whereas here there is only a single-shot setting. It would be good to use different symbols for the revenue/welfare objectives. Line 61: instead of $V$ use something that denotes a set of samples. Line 104: what is the value-to-go? The proof sketch of Theorem 5 is a bit confusing. Lines 157-158 "single item" is repeated. The notation $r_i, \hat{r}_i, r^*_i$ in Section 2.1 is a bit confusing. In Section 2.2 you can add a quick comment that choosing a price randomly will not help. In Line 23 "know" appears twice. In Line 173 drop "a". It might be useful to include the definition of the pseudo-dimension at least in the appendix.

**Questions:**

-What settings other than single-item auctions can the results be extended to?

-For other types of distributions such as regular/MHR, can you derive optimal bounds?

-For the bound on the pseudo-dimension, can't you use a direct argument like Morgenstern and Roughgarden (2016) for $t$-level auctions? I am not sure I understand the need to go through the dual class.

-Can you comment a bit on the differences of your results/approach and [GHTZ'21]?

**Limitations:**

Addressed.

---

> ### Author Rebuttal · Authors · 2024-08-06
>
> Thank you for your thoughtful review and comments!
>
> - *The conceptual ideas of the work feel a bit incremental, given the long line of work in the setting of learning optimal auctions. The results are limited to a very simple setting of single item auctions.*
>
> We view the simplicity of single item auctions as a major plus of our work, and we were surprised that the sample complexity of welfare/revenue maximization even in this simple setting had not been settled before. For example, our Theorem 1 entirely removes any dependency on the number of bidders in the sample complexity for product distributions (whereas the previous best known bound was linear). In terms of conceptual ideas, we want to highlight that a key difference compared to prior work is that we want to analyze the dynamic program corresponding to the optimal posted-price mechanism. Indeed, our Theorems 1 and 2 (for product distributions) crucially exploit the structure of the dynamic program for our specific problem.
>
>
> - *The running time of the algorithm for correlated distributions is exponential.*
>
> The runtime is $(Tn(Tn)^{1+|\mathcal{S}|})$, i.e. exponential in $|\mathcal{S}|$.  This is because there are $1+|\mathcal{S}|$ prices to decide, each of which can take $Tn$ possible values (one for each realized value in the $T$ samples of length $n$), and evaluating each combination of prices over each of the $T$ samples takes runtime linear in $n$.
>
> We had in mind settings where $|\mathcal{S}|$ is a small constant, in which case the runtime of Empirical Risk Minimization is polynomial in $T$ and $n$, and our sample complexity is also not growing with $n$.  This is well motivated in e.g. business settings where you can only change the price at the start of the month; in fact, there is substantial literature studying pricing policies where you cannot change the price too often (e.g. Cheung et al. 2017 below).
>
> Cheung, Wang Chi, David Simchi-Levi, and He Wang. "Dynamic pricing and demand learning with limited price experimentation." Operations Research 65.6 (2017): 1722-1731.
>
> - *For example, I found the use of the word "policy" a bit confusing. Usually, this would imply some dynamic decision process, whereas here there is only a single-shot setting.*
>
> The terminology we adopt is that a policy is a specific prescription of the price to set at each time, and the goal of the (learning) algorithm is to decide on a policy from the samples.  Technically the policy here still needs to be dynamic — once the item is sold, it must change the prices to infinity.  So since the prophet inequality/dynamic pricing problems we study are intrinsically dynamic programming problems, we decided to use the word “policy” for generality, although we agree that we could have used a different word such as “price vector” instead.
>
> - *It would be good to use different symbols for the revenue/welfare objectives ... It might be useful to include the definition of the pseudo-dimension at least in the appendix.*
>
> Thanks for catching these typos and suggestions on exposition!  We will definitely incorporate them in the next update.
>
> Regarding line 104, the value-to-go is the expected value of the (optimal) dynamic programming solution, conditional on the item still being unsold at time i.  It is referring to the quantity $r*_i$ in Section 2.2.
>
> Regarding the proof sketch of Theorem 5, sorry for the brevity.  We were trying to communicate the idea that statistically we need $\Omega(1/\epsilon^2)$ samples to get an $\epsilon$-approximation; however, under our construction, only 1/(1+|S|) fraction of the samples would be relevant.  Therefore, in order to get $\Omega(1/\epsilon^2)$ relevant samples we now need $\Omega((1+|S|)/\epsilon^2)$ samples to begin with, as formally shown in Theorem 5.
>
> - *What settings other than single-item auctions can the results be extended to?*
>
> A natural next question is to extend our results to the problem of posted pricing for selling $k$ identical items (instead of one). We believe that many of our techniques can be extended to this more general setting, but we focus on the single item setting since prior to our work even that was not understood properly.
>
>
> - *For other types of distributions such as regular/MHR, can you derive optimal bounds?*
>
> Thanks for the interesting question! Our upper bounds in Theorems 1 and 4 are already tight (up to constants) without needing to make additional assumptions such as MHR.  It is true that our lower bound constructions require distributions that are not MHR, and it is plausible to us that our lower bound construction from Theorem 2 (showing $\Omega(n/\epsilon^2)$ for product distributions) would not be possible when restricted to MHR.  Put another way, it is plausible that an $O(1/\epsilon^2)$ upper bound is possible even for revenue maximization on product distributions under regular/MHR valuations, a question we leave to future work.
>
> - *For the bound on the pseudo-dimension, can't you use a direct argument like Morgenstern and Roughgarden (2016) for t-level auctions? I am not sure I understand the need to go through the dual class.*
>
> Thank you for this question! It is indeed possible to directly bound the pseudo-dimension using first principles. In fact, that was our initial approach. However, we later realized we could leverage the existing result of Balcan et al., which resulted in a shorter and cleaner proof. This is why we opted to present it this way in our paper.
>
> - *Can you comment a bit on the differences of your results/approach and [GHTZ'21]?*
>
> The primary differences are: (1) the bounds in [GHTZ'21] are only applicable to product distributions; (2) even within the context of welfare maximization for product distributions, our work removes the dependency on number of bidders $n$ in their $O(n/\epsilon^2)$ bound, as shown in Theorem 1.

---

> > ### Comment · Reviewer_U8bp · 2024-08-09
> >
> > I would like to thank the authors for their thorough response. After reading their comments and the rest of the reviews, I've decided to increase my score to 6.

---

### Official Review · Reviewer_gD9M · 2024-07-15

**Soundness:** 4
**Presentation:** 2
**Contribution:** 3
**Rating:** 6
**Confidence:** 4

**Summary:**

The paper studies a setup of a series posted-price auctions that tries to sell a single item up to the first price acceptance. The authors seeking for the number of samples from buyer value distributions (sample complexity) to be able to setup near-optimal posted-price in the auctions. They consider two objectives (social welfare and revenue maximization) and different dependencies of buyer distributions (independent and correlated). The paper contributes proven both upper and lower on the sample complexity for the all four settings: the bounds are close up to logarithmic factors.

**Strengths:**

-	Clear math contribution

-	Well-structured proofs

**Weaknesses:**

-	No conclusions: the paper does not have a concluding discussion

-	No experimental evidence: asymptotic bounds are clear measures for growing (shrinking) parameters, but, for practice, constant factors are more important. It would be nice to understand how they work in practical cases.

**Questions:**

-	What are the factors (multiplicative constants) in the upper and lower bounds in Theorems 1-4?  What is behind O(...) and \Omega(...)?

**Limitations:**

the authors adequately addressed the limitations

---

> ### Author Rebuttal · Authors · 2024-08-06
>
> Thank you for your thoughtful review and comments!
>
> - *What are the factors (multiplicative constants) in the upper and lower bounds in Theorems 1-4? What is behind O(...) and \Omega(...)?*
>
> We emphasize that because we have normalized valuations to lie in [0,1], the $O(\ldots)$ and $\Omega(\ldots)$ notation is not hiding any dependencies other than absolute numerical constants.
>
> We would like to stress that optimizing the constants and coefficients in the sample complexity is not the main focus of our work.  We are interested in whether the sample complexity grows with $n$, which in our mind is a more first-order dependence.  Our main finding is that for product distributions, the sample complexity of welfare maximization does not grow with $n$, whereas the sample complexity of revenue maximization does.
>
> - *No conclusions: the paper does not have a concluding discussion*
>
> A rough, intuitive conclusion from this finding is that the dynamic programming thresholds computed from empirical distributions may be more prone to overfitting for revenue maximization than for welfare maximization.  Put another way, more samples need to be collected for the revenue maximization problem in order to avoid overfitting. We will add a concluding discussion to the next version of the paper.

---

### Official Review · Reviewer_mFS4 · 2024-07-15

**Soundness:** 3
**Presentation:** 3
**Contribution:** 3
**Rating:** 7
**Confidence:** 4

**Summary:**

The paper considers the problem of statistically estimating the optimal posted price mechanism for selling one item to multiple buyers. Here, it is assumed that the buyers appear sequentially, a price is presented to each of them and the sale is completed if the price posted to them is lower than their valuation. The mechanism aims to optimize one of two targets -- the welfare, defined here as the value of the bidder who wins the auction and revenue of the auctioneer which is the price paid by the winning bidder. It is assumed that the valuations of the bidders are drawn from a distribution and the paper aims to analyze the sample complexity of determining near-optimal posted prices for each of these settings where it is assumed that the learner receives iid samples from the bid distributions. Depending on the degree of dependence between the valuations of the bidders, the paper considers two settings -- firstly, the independent setting where the values are determined independently for each bidder and the dependent setting where the valuations are correlated across the bidders.

In the independent setting, it is shown that there exists a separation between the sample complexities of welfare and revenue maximization. They show that essentially $1 / \epsilon^2$ samples suffice for welfare maximization whereas revenue maximization necessarily \emph{requires} $\Omega (n / \epsilon^2)$ samples where $n$ is the number of bidders and $\epsilon$ an additive error parameter. In this case, they obtain near-optimal characterization of the sample complexity while the upper and lower bounds from prior work are $n / \epsilon^2$ and $1 / \epsilon^2$ respectively. In contrast, when the bids are correlated, the paper shows that $\Omega (n / \epsilon^2)$ is \emph{required} for both revenue and welfare maximization. To partially address this pessimistic bound, the paper also considers the setting where the class of possible posted prices is restricted. The particular restriction considered in the paper is by restricting the change points of the price schedule; that is, the price is only allowed to change at $k$ points in the sequence of prices. Here, it is shown that sample complexities independent of $n$ are obtainable and instead only depend on $k$.

The proof of the upper bound independent of $n$ for welfare maximization in the independent valuation setting is quite interesting. Essentially, the paper analyzes the dynamic programming algorithm for welfare maximization. Observing that the (welfare of the) posted prices have a closed-form solution in terms of the welfare of the next step, the paper proves that the sub-optimality of the posted prices may be bounded by a sum of error terms which satisfy a backwards martingale structure. The use of standard Martingale concentration bounds then yield the required bound. This proof is elegant and appears to be novel. Unfortunately, such martingale structure does not hold in the presence of dependencies between the valuations. Instead, they adopt a classical uniform convergence based approach and show that the statistical complexity may be controlled by the pseudo-dimension of the policy class.

Overall, this is a nice paper that obtains nearly optimal statistical characterizations in several fundamental settings, making several interesting contributions. The proofs of the results, the welfare maximization for independent valuations in particular, are elegant and well-presented. My main concerns are with the assumptions underlying the learning problem. The paper assumes that one obtains independent samples from the \emph{valuations} of the bidders. It is not clear how such samples are obtained -- it is not clear how one may obtain such samples in practice. More exposition on this point would be helpful. Furthermore, it would be helpful if the authors could comment on alternative settings circumventing the $\Omega (n / \epsilon^2)$ sample complexities incurred by the paper -- perhaps, by bounding the degree of dependence between the bidders?

**Strengths:**

See main review

**Weaknesses:**

See main review

**Questions:**

See main review

**Limitations:**

See main review

---

> ### Author Rebuttal · Authors · 2024-08-06
>
> Thank you for your thoughtful review and comments!
>
> - *The paper assumes that one obtains independent samples from the valuations of the bidders. It is not clear how such samples are obtained -- it is not clear how one may obtain such samples in practice. More exposition on this point would be helpful.*
>
> Valuation samples can be obtained in practice through marketing research such as consumer surveys.  We would like to emphasize that this is a standard model; the long line of literature on auction design from samples (e.g. CR16, and other citations in our paper) and pricing from samples (e.g. Huang et al. 2015 below) all generally focus on valuation samples.  Although there are alternate models of information such as the purchase probability at a single price (e.g. Allouah et al. 2023 below), moment ambiguity sets (e.g. Wang et al. 2024 below), a difference is that these models do not consider the randomness caused by sampling.
>
> Huang, Zhiyi, Yishay Mansour, and Tim Roughgarden. "Making the most of your samples." Proceedings of the Sixteenth ACM Conference on Economics and Computation. 2015.
>
> Allouah, Amine, Achraf Bahamou, and Omar Besbes. "Optimal pricing with a single point." Management Science 69.10 (2023): 5866-5882.
>
> Wang, Shixin, Shaoxuan Liu, and Jiawei Zhang. "Minimax regret robust screening with moment information." Manufacturing & Service Operations Management 26.3 (2024): 992-1012.
>
> - *Furthermore, it would be helpful if the authors could comment on alternative settings circumventing the Ω(n/ϵ^2) sample complexities incurred by the paper -- perhaps, by bounding the degree of dependence between the bidders?*
>
> Thank you for your question. This is indeed a fascinating area for future research. Recent works have explored welfare and revenue maximization under models with bounded correlations, such as linear correlations [Immorlica, Singla, and Waggoner, EC 2020] and graphical correlations of Markov Random Fields [Cai and Oikonomou, EC 2021]. It would be interesting to investigate the sample complexity within these frameworks, as these specific forms of correlations may allow one to circumvent the $\Omega(n/\epsilon^2)$ sample complexity lower bound from our Theorem 3.

---

> > ### Comment · Reviewer_mFS4 · 2024-08-08
> >
> > Thank you for the response! It would be great if the authors could include the above discussion in subsequent versions of the paper. I will retain my current evaluation.

---

### Decision · Program_Chairs · 2024-09-25

**Decision:**

Accept (spotlight)

**Comment:**

Executive Summary

The paper examines the sample complexity of posted-price mechanisms, for both independent (product) distributions and correlated distributions, and both welfare and revenue objectives. The authors assume bounded distributions (normalized in [0,1]), and seek additive approximations to the best posted-price policy. More precisely, they seek approximations of the form "at least OPT - eps with probability at least 1-delta".

The paper shows that for product distributions, the sample complexity for welfare maximization is O(1/eps^2) (omitting the dependence of delta), matching a known lower bound. Importantly, this is independent of n, the number of buyers. At the same time, for the revenue objective, they establish a lower bound of OmegaTilde(n/eps^2), and thus establish a separation to the welfare case.

The paper also considers the correlated case. For the welfare objective they give a lower bound of OmegaTilde(n/eps^2), matching a upper bound from prior work (and establishing a formal separation from the corresponding independent case). They complement this with postive results for welfare and revenue maximization, when the algorithm is constrained to change prices only a few times.

Recommendation

Overall, this is a very strong NeurIPS submission, which I am very happy to recommend for acceptance (as a spotlight if possible). This positive view of the paper is also reflected in the comments and suggestions from the reviewers, which I encourage the authors to take into account when preparing the final version.

Additional Requests

It appears to me that there is a closely related literature, which would be a natural reference point in the discussion. Namely, prior work on sample-based prophet inequalities, such as the following two papers (and follow-up work). I think the paper would benefit from discussing the differences between the approach taken in this paper, and the approach taken in this prior work (e.g. bounded distributions vs general distributions, prophet benchmark vs optimal online policy, etc.)

Correa, Duetting, Fischer, Schewior. Prophet Inequalities for I.I.D. Random Variables from an Unknown Distribution. EC'19
https://dblp.org/rec/conf/ec/CorreaDFS19.html?view=bibtex

Rubinstein, Weinberg, Wang. Optimal Single-Choice Prophet Inequalities from Samples. ITCS'20
https://dblp.org/rec/conf/innovations/RubinsteinWW20.html?view=bibtex

Another stream of related work that I feel deserves discussion is the one explored in the following paper, which explores the (non-)robustness of algorithms for the prophet inequality problem to inaccuracies in the distributions.

Duetting and Kesselheim. Prophet Inequalities with Inaccurate Priors. EC'19
https://dblp.org/rec/conf/ec/DuttingK19.html?view=bibtex